# SUMOylation Potentiates ZIC Protein Activity to Influence Murine Neural Crest Cell Specification

**DOI:** 10.3390/ijms221910437

**Published:** 2021-09-28

**Authors:** Helen M. Bellchambers, Kristen S. Barratt, Koula E. M. Diamand, Ruth M. Arkell

**Affiliations:** Early Mammalian Development Laboratory, John Curtin School of Medical Research, The Australian National University, Canberra, ACT 2601, Australia; hbellcha@iu.edu (H.M.B.); kristen.barratt@anu.edu.au (K.S.B.); koula.diamand@anu.edu.au (K.E.M.D.)

**Keywords:** TCF, *Foxd3*, post-translational modification, transcription factor, co-factor, ZIC, mouse neural crest cell

## Abstract

The mechanisms of neural crest cell induction and specification are highly conserved among vertebrate model organisms, but how similar these mechanisms are in mammalian neural crest cell formation remains open to question. The zinc finger of the cerebellum 1 (ZIC1) transcription factor is considered a core component of the vertebrate gene regulatory network that specifies neural crest fate at the neural plate border. In mouse embryos, however, *Zic1* mutation does not cause neural crest defects. Instead, we and others have shown that murine *Zic2* and *Zic5* mutate to give a neural crest phenotype. Here, we extend this knowledge by demonstrating that murine *Zic3* is also required for, and co-operates with, *Zic2* and *Zic5* during mammalian neural crest specification. At the murine neural plate border (a region of high canonical WNT activity) ZIC2, ZIC3, and ZIC5 function as transcription factors to jointly activate the *Foxd3* specifier gene. This function is promoted by SUMOylation of the ZIC proteins at a conserved lysine immediately N-terminal of the ZIC zinc finger domain. In contrast, in the lateral regions of the neurectoderm (a region of low canonical WNT activity) basal ZIC proteins act as co-repressors of WNT/TCF-mediated transcription. Our work provides a mechanism by which mammalian neural crest specification is restricted to the neural plate border. Furthermore, given that WNT signaling and SUMOylation are also features of non-mammalian neural crest specification, it suggests that mammalian neural crest induction shares broad conservation, but altered molecular detail, with chicken, zebrafish, and *Xenopus* neural crest induction.

## 1. Introduction

Neural crest cells (NCCs) are a transient population of multipotent cells that arise at the border of the neural and non-neural ectoderm (known as the neural plate border) during neurulation. These cells undergo an epithelial-to-mesenchymal transition before migrating through the embryo, forming a diverse range of cell types such that the neural crest is often considered the fourth germ layer [1]. There is a general understanding of the signaling molecules and transcriptional cascade that drives neural crest specification; this is collectively called the neural crest gene regulatory network (GRN) [2,3,4]. Specifically, some of the major signaling pathways, including FGF, BMP, and WNT, act together to induce the expression of several genes, termed neural plate border genes. Though these genes collectively mark the neural plate border, their individual expression extends beyond this region. The neural plate border genes act in concert with the major signaling pathways to trigger expression of a second set of genes, termed the neural crest specifiers. These are the first markers of NCC fate and are believed to ultimately control NCC formation and behavior through regulating the expression of neural crest effector genes. 

Although the signaling cascade that directs neural crest specification is known, there is a lack of detailed understanding regarding the actions of specific molecules and transcription factors in directing neural crest development, particularly when comparing mammals to non-mammalian models [5]. For example, it is unclear if canonical WNT signaling plays the same role in the mouse as it does in the other model organisms used to study neural crest formation. Knockout or knockdown of canonical WNT ligands causes impaired neural crest induction phenotypes in *Xenopus*, chick, and zebrafish [6,7,8,9,10,11,12] but this phenotype is not replicated when the same genes are knocked out in the mouse [5,13,14,15,16,17,18]. Barriga et al. proposed several technical explanations for this difference including that (1) the wingless/integrated 1(*Wnt1*)-*Cre* line typically used to study neural crest in mice may not be targeting the earliest stages of neural crest formation and (2) there may be redundancy or compensation from other WNT ligands. In fact, ectopic activation of canonical WNT signaling in the anterior of the mouse embryo via knockout of the WNT antagonist dickkopf-related protein 1 (*Dkk1*) [19], or conditional inactivation of the WNT transcriptional repressor transcription factor 7 like 1 (*Tcf7l1*) [20], induces ectopic neural crest induction, suggesting canonical WNT signaling is still part of the mammalian neural crest GRN.

There is growing evidence that non-mammalian neural crest specification and differentiation is regulated by the post-translational modification SUMOylation. SUMOylation is the attachment of the SUMO (small ubiquitin-like modifier) protein to target proteins via an enzymatic cascade involving a dimeric E1 activating enzyme, a single E2 conjugating enzyme UBC9 (also known as UBE2I), and a limited number of E3 ligating enzymes. In the chick embryo, UBC9 is enriched in the neural folds where the future NCCs are located, and morpholino knockdown of UBC9 reduces the expression of several neural crest specifiers (including *SNAIL2*, *SOX9,* and *FOXD3*) without altering expression of neural plate border genes [21]. In chick and *Xenopus* embryos, the SUMOylation of neural plate border and specifier transcription factors such as PAX7, SOX9/Sox9, and SOX10/Sox10 is critical for NCC development [21,22,23,24]. Together this suggests an overarching role for SUMOylation for the regulation of neural crest.

We have recently shown that another protein associated with neural crest, ZIC5, is also regulated by SUMOylation [25]. *ZIC5* is a member of the *ZIC* (zinc finger of the cerebellum) gene family that encodes multifunctional transcriptional regulators. In several species, including *Xenopus* and chick, a different member of the *ZIC* family, *Zic1*, is known to be a neural plate border gene. In *Xenopus*, *Zic1* is believed to be of critical importance for formation of the neural crest, as co-expression of *Zic1* with *Pax3*, another neural plate border gene, is sufficient to activate expression of neural crest specifiers including *Snail* and *Foxd3* [26,27,28] and to cause an epithelial-to-mesenchymal transition in *Xenopus* ectoderm explants [26]. In chick, expression of *ZIC1* has been found to be critical for activation of a *FOXD3* enhancer that drives trunk neural crest specification [29]. In mice, however, *Zic1* expression is not detected during the time of NCC induction [30] and deletion of murine *Zic1* does not lead to a neural crest phenotype (reviewed in [31]). Other members of the murine ZIC family (*Zic2*, *Zic3,* and *Zic5*) are expressed in the neurectoderm during gastrulation and neurulation, including in the neural plate border region that is the site of neural crest specification [30,32,33]. Loss-of-function mutations in either *Zic2* or *Zic5* result in the depletion of NCCs and lead to neural crest-related phenotypes including ventral spotting and reduced cephalic neural crest tissues [25,34,35,36,37,38], suggesting the other *Zic* genes have acquired the role of neural plate border genes from *Zic1* in mammalian species.

The defining feature of the ZICs is a zinc finger domain, closely related to the zinc finger domains of the GLI, GLIS, and NKL protein families [34]. The zinc finger domain is thought to have several functions including interacting with DNA [39,40], interacting with proteins [41,42], and possibly localizing ZICs to the nucleus [43,44]. One of the functions of the zinc finger domain is to enable the ZICs to bind TCF proteins, the key transcriptional mediators of canonical WNT signaling. TCF proteins bind DNA, with the TCF binding site referred to as a WNT-response element (WRE). In the absence of WNT ligand, the β-catenin molecule is degraded in the cytoplasm, causing TCF proteins to complex in the nucleus with other transcription factors to repress transcription at WREs. When WNT ligand is present, cytoplasmic β-catenin is stabilized and enters the nucleus where it complexes with TCF proteins and activates transcription at WREs [45]. The ZIC proteins bind TCF proteins and function as a co-repressor to block transcription at WREs when overexpressed in human cells or *Xenopus* and zebrafish embryos [42,46,47].

Outside the zinc finger domain, the ZICs are more varied, but all vertebrate ZICs share the zinc finger N-terminally conserved (ZF-NC) domain, a small domain (14–21 amino acids) that is located immediately adjacent to the zinc finger domain [48]. The function of the ZF-NC is unknown; however, we recently showed it contains a highly conserved SUMOylation consensus sequence that is targeted for SUMOylation in response to canonical WNT signaling in ZIC5 [25]. Here, we further investigated the connection between WNT signaling, ZIC function, and SUMOylation during mammalian NCC induction and specification. Using cell-based assays, we found that ZIC proteins 1–4 are SUMOylated at a lysine within the highly conserved ZF-NC domain. Via a combination of compound mouse mutants and cell-based assays, we showed that the three neural plate border-expressed Zic genes (*Zic2*, *Zic3* and *Zic5*) act co-operatively to drive the expression of the *Foxd3* specifier gene, and that SUMOylation enhances ZIC2 and ZIC3s trans-activation at a *Foxd3* enhancer element. In contrast, SUMOylation decreases ZIC2 and ZIC3s TCF co-repressor activity, and we found that the previously demonstrated ability of the ZIC proteins to inhibit WNT-dependent TCF-mediated transcription varies with the level of WNT stimulation. In the presence of a medium level of WNT stimulation, each ZIC protein inhibits TCF-mediated transcription, but this ability is impaired or lost when WNT signaling activity is high. In the mouse embryo, a TCF-dependent GFP reporter indicates that canonical WNT-dependent transcription is active in a subset of cells within the dorsal neurectoderm at the time of murine NCC specification. The loss of *Zic3* function leads to an increase in the size of the TCF-dependent transcription domain in the neurectoderm, indicating that during normal development ZIC represses TCF-mediated transcription in this tissue. Additionally, we found that the proportion of SUMOylated ZIC protein increases in response to high levels of WNT signaling. Together the data suggest that ZIC proteins play multiple roles within the neurectoderm at the time of NCC specification. In the lateral neurectoderm (where WNT levels are relatively low), they repress TCF-dependent transcription. At the neural plate border, in the presence of high WNT signals, SUMOylation of ZIC protein shifts ZIC activity from TCF co-repressor at WREs to a transcriptional activator of ZREs, including those that control *Foxd3* expression. Considering these results, we propose a model in which canonical WNT signaling changes the activity of ZIC protein at the neural plate border to cause localized NCC specification.

## 2. Results

### 2.1. The ZIC Proteins Can Be SUMOylated at a Conserved Lysine in the ZF-NC Domain

During murine NCC formation, ZIC5 regulation of *Foxd3* expression requires post-translational modification of the ZIC5 protein by SUMOylation at a lysine within the highly conserved ZF-NC domain [25]. The paralogous ZIC3 lysine is also a SUMO substrate [49], and a high-throughput proteomic study [50] and search for high-probability consensus SUMOylation motifs within the ZIC proteins [25,51] suggest all human ZIC proteins can be SUMOylated. To determine if this is the case, the various ZICs were assessed using the UBC9 fusion-directed SUMOylation (UFDS) system. In this system the potential target protein and UBC9 are expressed as a fusion protein, thereby removing the need for SUMO ligases [52,53]. To utilize this system, HEK293T cells were transiently transfected with a UBC9-fused version of the V5 epitope-tagged wild-type human ZIC1-4 (V5-UBC9-ZIC1-WT, etc.) alone or with either emerald green fluorescent protein (EmGFP)-tagged wild-type SUMO1 (EmGFP-SUMO1-WT) or a SUMOylation-defective SUMO1 mutant (EmGFP-SUMO1-∆GG) [54]. Attachment of EmGFP-SUMO1-WT should increase the molecular mass of target proteins by approximately 40 kDa; thus, the cells were lysed and subjected to SDS-PAGE and western blotting (WB) to assess the protein size. As shown in Appendix A, in each case, additional heavier, ZIC-specific bands were detected in addition to the unmodified ZIC band only in the presence of wild-type SUMO1 protein, indicating that ZIC1, ZIC2, ZIC3, and ZIC4 can be SUMOylated. The UBC9 fusion could force the post-translational modification of a protein not normally SUMOylated. To exclude this possibility, the transfections were repeated using ZIC proteins without the UBC9 fusion (i.e., with V5 epitope-tagged ZIC). As before, higher-molecular-mass bands were observed only in the presence of EmGFP-tagged SUMO1, indicating all members of the ZIC family were SUMOylated (Appendix A). In each case, the molecular mass of the ZIC protein increased by more than 40 kDa, which is consistent with the attachment of multiple EmGFP-tagged SUMO1 molecules.

SUMOylation generally modifies a lysine within a motif of conserved residues (ΨKXE), enabling the identification of high-probability SUMO sites via sequence analysis. In addition to the highly conserved SUMOylation site in the ZF-NC domain within ZIC1-4, our previous analysis showed that human ZIC1 and ZIC3 also contain a second high-probability canonical SUMOylation site at the boundary of zinc fingers 3 and 4 [25] (Figure 1a). To determine which of the predicted SUMOylation sites within ZIC1-4 are *bona fide* sites of SUMO attachment in HEK293T cells, each putative, modified lysine (K) was converted to an arginine (R) (which cannot be modified be SUMOylation) in the V5 epitope-tagged mammalian expression construct. Each construct was then transiently transfected into HEK293T cells in the presence or absence of EmGFP-SUMO1-WT before the cells were lysed and the resulting ZIC protein forms were analyzed by SDS-PAGE and WB. As shown in Figure 1b–e, for each ZIC protein, substitution of the ZF-NC lysine (i.e., ZIC1-K222R, ZIC2-K253R, ZIC3-K248R, and ZIC4-K125R) was sufficient to prevent all SUMO conjugation. In contrast, substitution of the lysine at the boundary of zinc fingers 3 and 4 (ZIC1-K333R of ZIC3-K359R) did not alter the ZICs’ SUMOylation status relative to the wild-type forms of these proteins. Even when the pool of SUMOylated ZIC was specifically increased by UBC9 fusion (using the UFDS system), only mutation of the ZF-NC lysine caused significant loss of the relevant-sized products (Appendix A). The data indicate that, in HEK293T cells, each ZIC protein is poly-SUMOylated at a canonical SUMOylation site within the ZIC ZF-NC domain. 

### 2.2. The Zic Genes Cooperate during Murine Neural Crest Cell Specification

To determine if, as for murine ZIC5 [25], SUMOylation potentiates ZIC transcription factor activity during NCC formation, we first identified those ZIC family members that are required for this process. During murine embryogenesis, three Zic genes (*Zic2*, *Zic3,* and *Zic5*) are expressed in the gastrula ectoderm fated to give rise to the neural crest and at the neural plate border [30,32]. In contrast, *Zic1* and *Zic4* are not expressed during gastrulation and early neurulation [30,34]. Neural crest defects have been reported in murine embryos with severe loss-of-function mutations in *Zic2* [36] or *Zic5* [38]. In each case, neural crest specification is impaired, resulting in decreased (but not absent) formation of NCCs. To determine whether *Zic2*, *3,* and *5* may act redundantly at the neural plate border, the expression of the *Foxd3* specifier gene, a direct target of *ZIC1* during chick NCC specification [29], was examined in murine embryos with decreased *Zic* gene dosage (Figure 2). Reduction of *Zic* gene dosage was achieved using the previously described null mouse strains: *Zic2^Ku^* [36], *Zic3^Ka^* [55,56], and *Zic5^-^* [25,38]. Consistent with previous studies, embryos homozygous for these alleles of *Zic2* or *Zic5* alone showed depletion of *Foxd3* expression in pre-migratory cranial and trunk neural crest (Figure 2a,b,d) relative to their stage-matched, wild-type littermates [25,35,36]. Similarly, embryos deficient in *Zic3* alone showed depletion of *Foxd3* (Figure 2a,c) demonstrating that all three of the murine *Zic* genes expressed at the neural plate border play a role in NCC specification and the control of *Foxd3* expression. To further reduce *Zic* gene dosage, the single mutant strains were crossbred. Compound alteration of *Zic2*/*Zic5* gene dosage is not possible by crossbreeding due to their bi-gene arrangement [34]; instead, *Zic3* and either *Zic2* or *Zic5* compound heterozygous mutants were generated. As shown in Figure 2a,e,f, cranial neural crest specification showed a greater dependence on *Zic3* and *Zic5* gene function. Comparatively, when embryos that lack *Zic3* and have only one functioning copy of either *Zic2* or *Zic5* were generated, there was a substantial depletion (or complete loss) of *Foxd3* trunk expression and very little cranial *Foxd3* expression was retained (Figure 2a,g,h). Together the data indicate *Zic2*, *Zic**3,* and *Zic**5* act redundantly at the murine neural plate border to direct expression of *Foxd3*.

### 2.3. SUMOylation Promotes ZIC Trans-Activation of the Foxd3 Enhancer

Having confirmed ZIC2, ZIC3, and ZIC5 cooperate to promote neural crest development, cell-based assays were employed to assess how SUMO modification of these proteins affects this ability. To evaluate ZIC regulation of *Foxd3*, a luciferase reporter construct containing the murine genomic region equivalent to the previously identified ZIC-responsive chick *FOXD3* enhancer [25,29,57] was used. We previously demonstrated that ZIC trans-activation of this reporter construct requires ZIC DNA binding [25]; thus, ZIC proteins act as classical transcription factors at this element. The ability of basal versus SUMOylated ZIC proteins to stimulate reporter expression was quantified. Two methods were used to prevent SUMOylation of the exogenous ZIC proteins. First, to specifically inhibit ZIC SUMOylation we employed the V5 epitope-tagged ZIC expression constructs in which the ZF-NC target lysine had been converted to a non-modifiable arginine or, second, SUMOylation was universally inhibited via the co-expression of a dominant-negative UBC9 molecule (Flag-UBC9-C93S; [58]). As shown in Figure 3a,c, both ZIC2 and ZIC3 require SUMOylation for maximal activity at the *Foxd3* ZIC-responsive enhancer, since arginine substitution at the ZF-NC target lysine reduces reporter trans-activation. Moreover, in the presence of universal SUMO inhibition, the trans-activation activity of the wild-type ZIC2 or ZIC3 protein is reduced to that of the respective SUMO-incompetent form (ZIC2-K253R and ZIC3-K248R) (Figure 3b,d). This suggests that, as for the ZIC5 protein [25], SUMOylation promotes the ZIC-based trans-activation of the *Foxd3* reporter construct. Conversely, the SUMO-incompetent forms of ZIC1 and ZIC4 were as effective as their wild-type counterparts at trans-activation of the *Foxd3* reporter construct when overexpressed in HEK293T cells (Appendix A).

### 2.4. The ZIC Proteins Are Context-Dependent Inhibitors of Canonical WNT-Dependent Transcription In Vitro and In Vivo

The activation of the *Foxd3* enhancer provides one mechanism by which ZIC proteins can regulate neural crest specification. Several studies have, however, demonstrated that ZIC proteins can act as transcriptional inhibitors of canonical WNT signals [42,46,47,56] via physical interaction with TCF proteins (which bind and repress or activate transcription at WREs). Given canonical WNT signaling can influence neural crest specification, such repression by the ZIC proteins could also influence specification of the neural crest. Thus, to further examine characteristics of this activity, a modified TOPflash/FOPflash vector system optimized for use with ZIC proteins was constructed. ZIC proteins are known to stimulate a variety of basal promoters widely used in heterologous reporter systems [40,56,59,60,61] including the TK promoter used in commercial TOPflash/FOPflash vectors. To circumvent this problem, new reporter constructs were generated (TOPflash equivalent: pGL4.20-β-globin-WREx3, and FOPflash equivalent with mutated WREs: pGL4.20-β-globin-MREx3) using a backbone, luciferase cDNA, and promoter (β-globin), each of which has been independently tested and shown to be neutral with respect to exogenous ZIC protein activity in HEK293T cells [57]. Canonical WNT signaling is constrained in HEK293T cells and can be stimulated by transfection of a β-catenin expression plasmid, with transcription at TCF sites driven by endogenous TCF proteins. The level of WNT activity in this system can be titrated via the use of alternative forms of β-catenin, such as a truncated and, thus, stabilized form of β-catenin (β-catenin-ΔN89 [62]), which drives higher luciferase activity (Appendix A). To measure the relative ability of all mammalian ZIC proteins to inhibit WNT-dependent transcription, each protein was co-transfected with the full-length form of β-catenin (V5-β-CAT) or the stabilized form (ΔN89-β-CAT) and luciferase activity of pGL4.2-β-globin-WREx3 quantified (Figure 4a,b,c). ZIC inhibition of TCF-dependent transcription was greatest in the presence of wild-type β-catenin, and, for each ZIC protein, transcription inhibition was decreased (ZIC1 and 2) or ablated (ZIC3, 4, and 5) in the presence of β-catenin (Figure 2c). Luciferase transcription in these assays was dependent upon the presence of the TCF binding sites (i.e., the WRE) in the reporter construct since transcription was reduced to background levels when the binding sites were mutated (i.e., the MRE; Appendix A).

The results imply that, in HEK293T cells, ZIC proteins can inhibit TCF-dependent transcription at low levels of WNT activity, but that ZIC transcription inhibition is overcome when WNT activity increases. To test whether ZIC proteins influence TCF-dependent transcription in vivo, a WNT-reporter mouse strain (TCF/Lef:H2B-GFP; MGI: 4881498) was examined. This transgenic strain incorporates a human histone H2B-green fluorescent protein (GFP) fusion protein whose expression is driven by six repeats of the same WRE employed in the luciferase assays [63] followed by a hsp68 minimal promoter. In wild-type mouse embryos *Zic2*, *3,* and *5* are all expressed throughout the neurectoderm but, beginning at the early somite stages, their transcripts disappear from the future ventral neurectoderm and become increasingly restricted to the future dorsal neurectoderm [34]. Thus, at the time of NCC specification the genes are expressed in a dorsal domain of the neurectoderm, which is broader than the crest-forming region. In wild-type embryos, the GFP reporter reveals WNT-dependent transcription restricted to the neural plate border, as expected. When crossed with mice null for *Zic3*, embryos that lack *Zic3* function (*Zic3^Ka/Y^*) showed an increase in the domain of GFP reporter activity in regions of normal *Zic3* expression, including at the neural plate border and within the lateral neurectoderm at the time and site of neural crest cell specification (Figure 4e–h). This indicates that *Zic* expression is required for repression of TCF-dependent transcription in the lateral neurectoderm (but not neural plate border) and is consistent with the model from the cell-based studies, which implies that ZIC proteins are context-dependent inhibitors of canonical WNT/β-catenin driven transcription.

### 2.5. SUMOylation Decreases ZIC1, ZIC2, and ZIC3 Inhibition of TCF-Dependent Transcription

We next evaluated how SUMOylation affects the ZIC modification of TCF-dependent transcription as measured by the ZIC-optimized WRE-reporter system in HEK293T cells. To take account of the different dose-dependent inhibitory abilities of ZIC2 and ZIC3 (see Figure 4), different β-catenin expression constructs were used to stimulate TCF-dependent transcription for different ZIC proteins. The role of SUMOylation was evaluated both by specific inhibition of ZIC SUMOylation through expression of SUMO-incompetent ZIC proteins (V5-ZIC2-K253R or V5-ZIC3-K248R) or universally by co-expression of Flag-UBC9-C93S with V5-ZIC2-WT or V5-ZIC3-WT. For both ZIC2 and ZIC3, overexpression of the SUMO-incompetent forms significantly enhanced suppression of TCF-dependent transcription, relative to the respective wild-type protein (Figure 5a,a’,c,c’). Furthermore, the presence of dominant-negative UBC9 converted wild-type ZIC2 or ZIC3 into a stronger inhibitor of TCF-dependent transcription (Figure 5b,b’,d,d’), indicating that SUMOylation of the ZF-NC lysine accounts for the increased inhibitory activity of the K-R ZIC variant proteins. When the non-neural plate border ZIC proteins were assayed in this manner, it was found that SUMOylation of ZIC1 similarly enhanced its capacity to inhibit TCF-dependent transcription, but that SUMOylation did not alter ZIC4 suppression activity (Appendix A).

### 2.6. ZIC Protein SUMOylation Is Enhanced in a High-WNT Environment

Murine NCC specification occurs at a region of high canonical WNT activity (see Figure 4e,e’,g,g’), and we previously showed that the proportion of SUMOylated ZIC5 protein increases in HEK293T cells following stimulation of WNT activity. This theoretically provides a mechanism to alter the balance of ZIC transcription factor versus co-factor activity within the neurectoderm dependent upon the level of canonical WNT activity at a given location. To determine whether the other ZIC proteins respond similarly to WNT activity, the proportion of SUMOylated ZIC protein was compared between HEK293T cells at basal state and following treatment with the potent GSK3 inhibitor (2′Z,3′E)-6-Bromoindirubin-3′-oxime (BIO) using a protocol known to stimulate canonical WNT signaling in HEK293T cells [64]. For this experiment we selected one neural plate border ZIC protein (ZIC2) and one non-neural plate border ZIC protein (ZIC1), to determine if they responded differentially. As shown in Figure 6a–d, the proportion of SUMOylated ZIC1 and ZIC2 protein is substantially increased following BIO treatment, indicating that canonical WNT signaling can regulate the post-translational modification of these proteins.

## 3. Discussion

Here we demonstrated that, during normal development, murine ZIC proteins play two distinct roles during NCC specification. First, in the lateral neurectoderm (a region of relatively low WNT activity), ZIC3 (and potentially ZIC2 and ZIC5) acts at WREs to repress the expression of WNT target genes. Second, at the neural plate border (a region of high WNT activity and where WREs lead to gene transcription), ZIC proteins cooperate to drive the expression of the *Foxd3* NCC specifier gene, likely via direct binding to and trans-activation of ZREs. The ability of the ZIC proteins to switch from co-repressor at WREs to transcription factor at ZREs in a high-WNT environment can be facilitated by ZIC-protein SUMOylation, in response to the high level of WNT signaling. Together these data are synthesized into a working model of the role of ZIC proteins during NCC specification in Figure 7. In support of this model, we recently showed that ZIC5 SUMOylation is essential during murine development to drive optimal expression of *Foxd3* and specification of the neural crest [25].

We previously showed that *Foxd3* expression during murine NCC specification requires *Zic5* SUMOylation at a lysine within the deeply conserved ZF-NC protein domain [25]. We now show that each of the ZIC proteins can be polySUMOylated at the lysine within a high-probability SUMOylation consensus sequence in the ZF-NC domain and that this is the sole site of their SUMOylation. This confirms and extends the work of other studies suggesting ZIC proteins are a SUMO substrate [49,50] and supports the need for SUMOylation as a driver of the evolutionary conservation of the ZIC ZF-NC domain [25]. Furthermore, as for ZIC5, the steady-state level of ZIC1 and ZIC2 SUMOylation increases in the presence of high WNT activity. Although we have not directly tested ZIC3 and ZIC4 in this assay, it is possible that the SUMOylation of one, or both, of these proteins is also increased in a high-WNT environment. For the neural plate border *Zic* genes (*Zic2*, *Zic3,* and *Zic5*), our work here and in Ali et al., 2021, show that ZIC SUMOylation consistently correlates with decreased TCF co-repressor activity at WREs and increased trans-activation at the ZREs in the *Foxd3* enhancer. As previously described, this SUMOylation-driven switch in ZIC function extends the reach of WNT signaling, enabling a high-WNT environment to activate transcription not only at WREs but also at ZREs [25], and it would be interesting to determine if other genes involved in neural crest specification are also regulated by this process. The work presented here also confirms that the well-described requirement for SUMOylation during NCC specification in chick and *Xenopus* embryos [21,22] is conserved in mammals. As with other features, however, the molecular details appear to have altered. For example, mRNA transcripts and protein levels of the SUMO E2 ligase UBC9, which is essential for SUMO conjugation, are enriched in the neural folds of chick where the future NCCs are located [21], whereas the same enrichment of SUMO pathway molecules is not seen in mouse embryos [65].

The work presented here confirms previous studies that indicate that the ZIC proteins contribute to transcription control downstream of WNT signaling. Several gain-of-function studies show that the ZIC proteins can directly bind TCF7L2 and, when co-expressed with TCF7L2, can inhibit TCF-dependent transcription at WREs [42,46,47,56] in cells and in *Xenopus* and zebrafish embryos. Using a ZIC-optimized reporter assay in HEK293T cells, and various levels of WNT activation, we showed here that ZIC repression of WREs is overcome in high-WNT environments and that ZIC3, ZIC4, and ZIC5 are the most sensitive to this effect. Furthermore, we demonstrated for the first time that in vivo loss of ZIC activity (in murine *Zic3* null embryos) leads to an increase in TCF-dependent transcription at WREs, confirming that ZIC co-repression at WREs is required for normal development. It would be interesting to conduct similar experiments with other murine *Zic* alleles to determine when and where during embryonic development the various ZIC proteins repress canonical WNT signaling. The work also showed that the WNT signaling pathway is active at the right time and place to be involved in the earliest steps of murine NCC development and that altered WNT activity is associated with a NCC phenotype. This supports a conserved requirement for WNT signaling in mammalian NCC formation and suggests there is merit in attempting new strategies to identify the relevant murine WNT ligands and pathway molecules.

Previous studies of the role of *Zic* genes at the neural plate border showed the importance of *Zic1* in non-mammalian model organisms [26,27,28,29]. Here we showed that, during evolution, the role of ZIC in NCC specification has transferred and been split between multiple *Zic* genes, such that, in the mouse, *Zic2*, *Zic3,* and *Zic5* together control NCC development. We showed these genes have adopted the function of activating *Foxd3* expression in the dorsal neurectoderm during NCC specification. We previously showed that ZIC5 can trans-activate an evolutionarily conserved enhancer sequence containing ZREs that was first identified in chick embryos [29] in a DNA binding-dependent manner [25]. Here we showed that both ZIC2 and ZIC3 trans-activate this enhancer element and that their SUMOylation enhances this ability. It remains possible that, in vivo, *Foxd3* expression is activated by default in the dorsal neurectoderm where ZIC activity as a TCF co-repressor is lost due to high WNT signaling. Such a scenario would, however, be expected to lead to greater *Foxd3* expression in the *Zic3* null embryos due to the enlarged size of the TCF-dependent expression domain. The finding of decreased *Foxd3* expression in the dorsal neurectoderm of *Zic3* null embryos argues that ZIC3 directly drives trans-activation of *Foxd3* expression during NCC specification. Although it seems likely that the murine *Zic* genes would play a broader role in NCC specification by co-operatively regulating other genes at ZREs, we did not specifically test this here. Additionally, a more complete understanding of the cooperative role of the *Zic* genes is hindered by the mid-late-gestation lethality of *Zic* null alleles (which occurs at earlier embryonic stages in *Zic* compound mutants [66]) and the awaited development of additional mouse reagents such as conditional alleles of *Zic2* and *Zic5*.

The balance between non-neural and neural fates is thought to be driven by WNT signaling levels [67]. SUMOylation of ZIC proteins in a high-WNT environment can explain how NCCs arise at a discrete region of the developing neurectoderm, despite a broader expression domain of the ZIC neural plate border transcription factors and active WNT signaling in the non-neural ectoderm. In regions of lateral neurectoderm, where WNT signaling is relatively low, the ZIC proteins predominantly act as TCF co-repressors at WREs and are not available to bind ZREs. At the neural plate border, high-WNT activity causes ZIC SUMOylation, preferencing their activity at ZREs and enabling TCF to complex with β-catenin and activate WREs. Further laterally still, in the non-neural ectoderm where ZIC proteins are not expressed, TCF/β-catenin activity at WREs will occur, in the absence of ZIC binding to ZREs. It would be interesting to determine whether ZIC SUMOylation is used at other times and places during development to resolve WNT activity gradients.

## 4. Materials and Methods

### 4.1. Mouse Strains and Husbandry

Mice were maintained according to Australian Standards for Animal Care under protocol A2018/36 approved by The Australian National University Animal Ethics and Experimentation Committee for this study. The *Zic5^tm1Sia^* targeted null allele (MGI:3574814) of *Zic5* (*Zic5^-^*) [38] was backcrossed for 10 generations to the C3H/HeH inbred strain and subsequently for >10 generations to the C57BL6/J inbred strain. The kumba (*Ku*) allele of *Zic2* [36,68] was maintained by continuous backcross to C3H/HeH mice. The katun (*Ka*) allele of *Zic3* [56] and the TCF/Lef:H2B-GFP transgenic strain (MGI: 4881498, [63]) were maintained by continuous backcross to C57BL/6J inbred mice. Mice were maintained in a light cycle of 12 h light:12 h dark, the midpoint of the dark cycle being 12 a.m.. Embryos were dissected from 49–150-day-old pregnant females. Mice and embryos were genotyped by PCR screening of genomic DNA extracted from ear biopsy tissue or embryonic tissue, respectively [69]. For the *Zic5^-^*, *Zic3^Ka^*, and TCF/Lef:H2B-GFP strains, genomic DNA (50 ng) was amplified for High-Resolution Melt Analysis (HRMA) using IMMOLASE DNA Polymerase using the primers and PCR conditions previously described: *Zic5^-^* [25,70], *Zic3^Ka^* [56,70], and TCF/Lef:H2B-GFP [63]. For the *Zic2^Ku^* strain, genomic DNA (50 ng) was amplified for Allelic Discrimination using the primers and PCR conditions previously described [36].

### 4.2. Embryo Collection and Pre-Processing

All embryos were dissected from maternal tissue and Reichert’s membrane removed in 10% (v/v) fetal bovine serum in phosphate buffered saline (PBS) as described previously [71]. Embryos were staged according to Downs and Davies [72]. Embryos for whole mount in situ hybridization (WMISH) were transferred to 4% paraformaldehyde (PFA; Millipore Sigma, St. Louis, MO, USA: P6148) in 1X PBS and fixed overnight at 4 °C, before being used immediately or dehydrated via a methanol series and stored in 100% methanol at –20 °C until required. Embryos for whole mount immunofluorescence (WMIF) were transferred to 4% paraformaldehyde (PFA; Millipore Sigma: P6148) in 1X PBS and fixed overnight at 4 °C, before being used immediately or stored in 1X PBS at 4 °C.

### 4.3. Whole Mount in Situ Hybridization

Whole embryos were re-hydrated using standard procedures [71] and rinsed in PBT (PBS with 0.1% Tween-20 [Millipore Sigma: P9416]). WMISH to *Foxd3* was performed as previously described [25,36,71,73]. A minimum of four, eight somite-stage embryos per genotype were compared to precisely stage-matched, wild-type littermates. Upon completion of the WMISH procedure, embryos were postfixed in 4% PFA and transferred via a glycerol series to 100% glycerol. For photography, embryos were flat-mounted under a glass coverslip and photographed in a Nikon SMZ 21500 Stereomicroscope and DS-Ri1 camera (Nikon Inc., Melville, NY, USA).

### 4.4. Whole Mount Immunofluorescence and Confocal Microscopy

Whole embryos were rinsed in PBT and permeabilized in 1% H_2_O_2_ [Millipore Sigma: H1009] for 10 min at room temperature (RT) followed by 10 mg/mL proteinase K [Roche, Basel, Switzerland: 3115879001] in PBT for 5 min, RT, and 2 mg/mL glycine in PBT for 5 min, RT. For GFP detection, embryos were blocked in 5% skim milk powder (w/v; Diploma, New Zealand Dairy Products Bangladesh Ltd, Dhaka, Bangladesh: 3001742) in PBT at RT for a minimum of 3 h, incubated overnight at 4 °C with α-GFP (Abcam, Cambridge, UK; cat. no. ab6673; 1:100) primary antibody, blocked again in 5% skim milk powder at RT for a minimum of 3 h, and incubated for 48 h at 4 °C with a donkey α-goat Alexa Fluor Plus 647 (ThermoFisher Scientific, Waltham, MA, USA; cat. no. A32849; 1:500) secondary antibody. Embryos were rinsed in 5% skim milk powder followed by PBT at RT before incubation with DAPI (0.005 mg/mL in PBS; ThermoFisher Scientific, cat. no. D3571; 1:1000) for 10 min at RT. Whole embryos were taken through a glycerol series [71] and mounted on SuperFrost Plus microscope slides or single-well cavity slides (ProSciTech, Kirwan, QLD, Australia: G341, 1.25 mm) and imaged through glass coverslips.

Laser scanning confocal data of immunofluorescent embryos and cryosections were acquired using a Zeiss LSM800 with Airyscan Super-resolution confocal microscope with a 10X and 20X objectives (Zeiss, Oberkochen, Germany) and DIC optics. Fluorophores were excited using a 405-nm diode laser (Hoechst-33342; DAPI) or 633-nm HeNe laser (AlexaFluor-633/647). Confocal images were acquired using ZEN BLUE as *z* stacks of *xy* images taken at 3-μm *z*-intervals for whole embryos and 0.5-μm for cryosections. A minimum of six embryos per genotype per stage were compared to precisely stage-matched, wild-type littermates. Maximum Intensity Projection 3D reconstructions of confocal z-stacks were created for each embryo. Images were processed and prepared using Adobe Photoshop 2021 (Adobe, San Francisco, CA, USA).

Post-whole mount confocal microscopy, embryos to be cryosectioned were taken through a series of Tissue-Tek OCT (ProSciTech: IA018) cryoembedding solution dilutions in 30% sucrose/PBS (1:3 OCT:30% sucrose, 1:1 OCT:30% sucrose, 3:1 OCT:30% sucrose) before being embedded in molds in 100% OCT and frozen on dry ice. OCT blocks were sectioned using ThermoFisher Scientific; HM525 NX Cryostat at 10–16 μm and the sections were adhered to SuperFrost Plus microscope slides (ThermoFisher Scientific: 4951PLUS4) coated in 2% (3-Aminopropyl)triethoxysilane (Millipore Sigma: A3648) diluted in acetone to ensure section adherence. Slides were stored at –20 °C and subjected to confocal microscopy as above.

### 4.5. Plasmids

The generation of pENTR3C-ZIC2-WT, pENTR3C-ZIC3-WT, pENTR3C-ZIC5-WT, V5-ZIC2, V5-ZIC3, V5-ZIC5, V5-DEST, and ΔN89-β-cat has been described previously [56], as has the generation of V5-β-catenin, pGL4.2-β-globin-Foxd3, EmGFP-SUMO-wt, EmGFP-SUMO-ΔGG, pSG5-HA-hUBC9, and FLAG-UBC9-C93S [25]. The generation of pENTR3C-ZIC1, pENTR3C-ZIC4, V5-ZIC1-wt, and V5-ZIC4-wt has also been described previously [57].

The 3× optimal TCF binding sites (WNT response elements; WRE) or 3× mutant TCF binding sites (mutant WNT response elements; MRE) were inserted into KpnI/HindIII restriction binding sites of pGL4.2 to generate pGL4.2-WREx3 and pGL4.2-MREx3. The human β-globin minimal promoter was amplified from pGL4.2-β-globin and inserted into HindIII restriction enzyme site of pGL4.2-WREx3 and pGL4.2-MREx3 to generate pGL4.2-β-globin-WREx3 and pGL4.2-β-globin-MREx3.

Specific point mutations were introduced into pENTR3C-ZIC3 using the QuikChange II Site-Directed Mutagenesis kit (Agilent, Santa Clara, CA, USA) to generate pENTR3C-ZIC3K248R (using primers Ark1181_F 5′- ATGCGGCAGCCTATCAGGCAGGAGCTGTCG-3′ and Ark1182_R 5′-CGACAGCTCCTGCCTGATAGGCTGCCGCAT-3′) and pENTR3C-ZIC3-K259R (using primers Ark1183_F 5′-GAGAAACCTTTCAGATGTGAATTTGAAGGC-3′ and Ark1181_R 5′- GCCTTCAAATTCACATCTGAAAGGTTTCTC-3′). Specific point mutations were introduced into pENTR3C-ZIC1 and pENTR3C-ZIC2 via overlap extension PCR to generate pENTR3C-ZIC1-K333R (using mutagenesis primers Ark1470_F 5′-AGCCCTTCAGGTGCGAGTT-3′ and Ark1471_R 5′-AACTCGCACCTGAAGGGCT-3′ and primers at the ends of ZIC1 Ark1460_F 5′-ATCCGGTACCGAATTCATGCTCCTGGACGCCGG-3′ and ZIC1 Ark1461_R 5′- GTGCGGCCGCGAATTCTTAAACGTACCATTCGTTAAAATTGG-3′) and pENTR3C-ZIC2-K253R (using mutagenesis primers Ark1230_F 5′-CGGCAGCAGTGCATCAGGCAGGAGCTAATC-3′ and Ark1231_R 5′-GATTAGCTCCTGCCTGATGCACTGCTGCCG-3′ and primers at the ends of ZIC2 Ark1168_F 5′- ATCCGGTACCGAATTCAGTGTGGTGGAATTCCTGGCC-3′ and Ark1150_R 5′- GTGCGGCCGCGAATTCGAGGGTTAGGGATAGGCTTAC-3). To generate pENTR3C-ZIC1-K222R and pENTR3C-ZIC4-K125R, a 5′ segment of each of pENTR3C-ZIC1 and pENTR3C-ZIC4 was amplified with primers containing the desired ZIC1-K222R and ZIC4-K125R mutations (using mutagenesis primer Ark1462_R 5′-GATGAGCTCTTGCCTGATGG-3′ and wild-type end primer Ark1463_F 5′-ATCCGGTACCGAATTCATG-3′ in the case of ZIC1-K222R and using mutagenesis primer Ark1428_R 5′-ATGAGCTCCTGTCTGATGG-3′ and wild-type end primer Ark1463_F 5′-ATCCGGTACCGAATTCATG-3′ in the case of ZIC4-K125R). These fragments were inserted into KpnI/SacI restriction sites of pENTR3C-ZIC1 and pENTR3C-ZIC4 to create pENTR3C-ZIC1K222R and pENTR3C-ZIC4K125R.

For all pENTR3C-ZIC constructs, UBC9 fusion constructs were generated by PCR amplifying UBC9 cDNA (with stop codon deleted) from pSG5-HA-hUBC9 [74] and inserting the resulting fragment into the KpnI restriction enzyme site at the N-terminus of each ZIC to create the various pENTR3C-UBC9-ZIC fusion constructs. For both the pENTR3C-ZIC constructs and the pENTR3C-UBC9-ZIC constructs, the ZIC sequence was transferred to the destination clone pcDNA3.1/nV5-DEST™ (ThermoFisher Scientific) via a Gateway^®^ LR Clonase reaction (as per manufacturer’s instructions; ThermoFisher Scientific) to produce the V5-ZIC and V5-UBC9-ZIC expression plasmids.

### 4.6. Cell Culture, Subcellular Fractionation, SDS-PAGE, and Western Blotting

The human embryonic kidney cell line (HEK293T) was cultured and transiently transfected as previously described [56]. Subcellular fractionation, SDS-PAGE, and WB were performed as described in [25].

### 4.7. SUMOylation Assays

For the UFDS experiments, HEK293T cells grown in six-well TC plates (approximately 1 × 10^6^ cells, Corning^®^, Corning NY, USA; Cat. no. CLS3516) were transfected with 2 µg ZIC with or without co-transfection of 2 µg of an EmGFP-SUMO expression construct. For non-UFDS experiments, 12 µg ZIC with or without co-transfection of 12 µg of an EmGFP-SUMO expression construct cells were grown in 100-mm TC dishes (approximately 7 × 10^6^ cells, Corning^®^; Cat. no. CLS430167). For experiments not including BIO ((2′Z,3′E)-6-Bromoindirubin-3′-oxime) treatment, 24 h after transfection, cells were lysed and analyzed via WB. For experiments including BIO treatment, 6 h after transfection, cells were dissociated from the growth surface using 0.05 g/L trypsin and replated into two 100-mm TC dishes. Then, 24 h after transfection, media was removed and replaced with either media including both 5 µM BIO and 0.5% DMSO or media with only the 0.5% DMSO. Then, 24 h after transfection, the cells were lysed and analyzed by WB. The relative amounts of SUMOylated ZIC with and without BIO treatment were quantified from scans of the resulting WBs using ImageJ (NIH).

### 4.8. Luciferase Reporter Assays

Luciferase assays were performed as described previously [25,56], but with different initial transfection conditions. For the ZIC2 and ZIC3 *Foxd3* reporter assays, HEK293T cells grown in 12-well plates were transfected with a total of 2 µg of DNA, comprised of 0.8 µg *Foxd3* reporter, 0.8 µg of either a ZIC expression construct or the empty construct (pcDNA3.1/nV5-DEST™), and 0.4 µg of either the empty construct (pcDNA3.1/nV5-DEST™) or the dominant negative SUMOylation inhibitor FLAG-UBC9-C93S. For the ZIC1 and ZIC4 *Foxd3* reporter assays, HEK293T cells grown in 12-well plates were transfected with a total of 1.6 µg of DNA, comprised of 0.8 µg *Foxd3* reporter and 0.8 µg of either a ZIC expression construct or the empty construct (pcDNA3.1/nV5-DEST™). For the *Z3M2* reporter assays, HEK293T cells grown in 12-well plates were transfected with a total of 2 µg of DNA, comprised of 0.8 µg *Z3M2* reporter, 0.8 µg of either a ZIC expression construct or the empty construct (pcDNA3.1/nV5-DEST™), and 0.4 µg of either the empty construct (pcDNA3.1/nV5-DEST™) or the dominant negative SUMOylation inhibitor FLAG-UBC9-C93S. For the WNT inhibition assays, HEK293T cells grown in six-well plates were transfected with a total of 4.5 µg of DNA, comprised of 1.5 µg pGL4.2-β-globin-TCFx3 reporter, 1.5 µg β-catenin expression construct (either V5-β-catenin or ΔN89-β-catenin), 1 µg of either a ZIC expression construct or the empty construct (pcDNA3.1/nV5-DEST™), and 0.5 µg of either the empty construct (pcDNA3.1/nV5-DEST™) or the dominant negative SUMOylation inhibitor FLAG-UBC9-C93S. In addition, to assess WNT background levels, for one well the β-catenin expression construct was substituted with the empty construct (pcDNA3.1/nV5-DEST™).

### 4.9. Antibodies

The following primary antibodies were used for WBs: mouse monoclonal anti-V5 (1:5000 dilution WB, ThermoFisher Scientific, cat. no. R960-25), rabbit polyclonal anti-GFP (1:1500 dilution WB, Cell Signaling, Danvers, MA, USA; cat. no. 2555), rabbit polyclonal anti-Lamin B1 (1:1500 dilution WB, Abcam, cat. no. ab16048), mouse monoclonal anti-β-tubulin (1:1000 dilution WB, Abcam, cat. no. ab7792), mouse monoclonal anti-TATA Binding Protein (TBP) (1:2000 dilution WB, Abcam, cat. no. ab818), goat polyclonal β-catenin C-18 (1:500 dilution WB, Santa Cruz Biotechnology, Dallas, TX, USA; cat. no. sc-1496), and mouse monoclonal anti-UBC9 (1:1000 dilution WB, BD Biosciences, Franklin Lakes, NJ, USA; cat no. 610748). Secondary antibodies used for WB (1:5000 dilution) were: horse radish peroxidase (HRP)-conjugated rabbit anti-mouse, rabbit anti-goat, and goat anti-rabbit (Zymed, ThermoFisher Scientific). All antibodies were diluted in blocking buffer (5% skim milk [Diploma], 0.02% Tween-20 in PBS except for the GFP antibody, for which the PBS was replaced with TBS [Tris-Buffered Saline; 50 mM Tris, 150 mM NaCl, pH 7.5]).

### 4.10. Statistical Analysis

For cell-based assays where one representative experiment is shown, the standard deviation was calculated from three internal repeats using Excel. For analysis of the pooled raw data (from a minimum of three external repeats) of these assays, GenStat software (VSN International, Hemel Hempstead, UK) was used to test for normality (W-test) and to perform a two-way ANOVA and Bonferroni multiple comparison test. Post-statistical analysis, the values calculated by the ANOVA and the SEM were normalized to the negative control and relative values plotted.

For statistical analysis of the amount of SUMOylated ZIC in response to BIO treatment, GenStat software was used to perform a paired two-sample *t*-test on data from three independent repeats. Post-analysis, predicted means and standard error of difference (SED) were plotted.

## Figures and Tables

**Figure 1 ijms-22-10437-f001:**
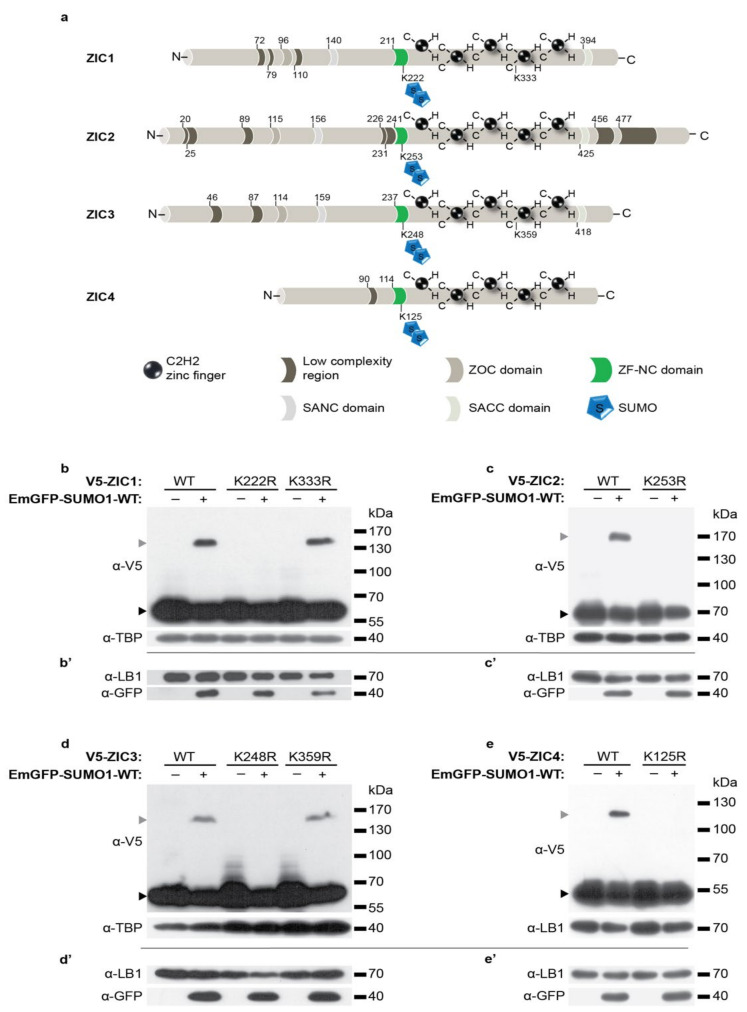
ZIC proteins are SUMOylated within the ZF-NC domain. (**a**) Schematic of human ZIC1-4 showing the SUMO attachment site. (**b**,**c**,**d**,**e**) Representative WBs of HEK293T cell nuclear fractions following transfection with the V5 epitope-tagged ZIC expression plasmid shown, with or without EmGFP-tagged SUMO1-WT expression construct. (**b**) ZIC1, (**c**) ZIC2, (**d**) ZIC3, and (**e**) ZIC4. The SUMO-dependent higher-molecular-mass form of each protein is not produced when the putative target lysine (K) within the ZF-NC (ZIC1 K222, ZIC2 K253, ZIC3 K248, and ZIC4 K125) is changed to an arginine (R), indicating these are the sole site of SUMO attachment in each protein. The higher-molecular-mass protein is produced when the other putative target lysines are altered (ZIC1 K333 and ZIC3 K359), indicating that these lysines are not a SUMO target. Gray arrow, SUMOylated ZIC; black arrow, basal ZIC. (**b**’,**c**’,**d**’,**e**’) WB to show overexpressed EmGFP-SUMO1-WT protein and corresponding loading control. For each protein, *n* = 3 independent transfections and WB. For WB of nuclear fractions (using antibodies against V5 or GFP), antibody against TBP or Lamin B1 (LB1) was used as a loading control, respectively.

**Figure 2 ijms-22-10437-f002:**
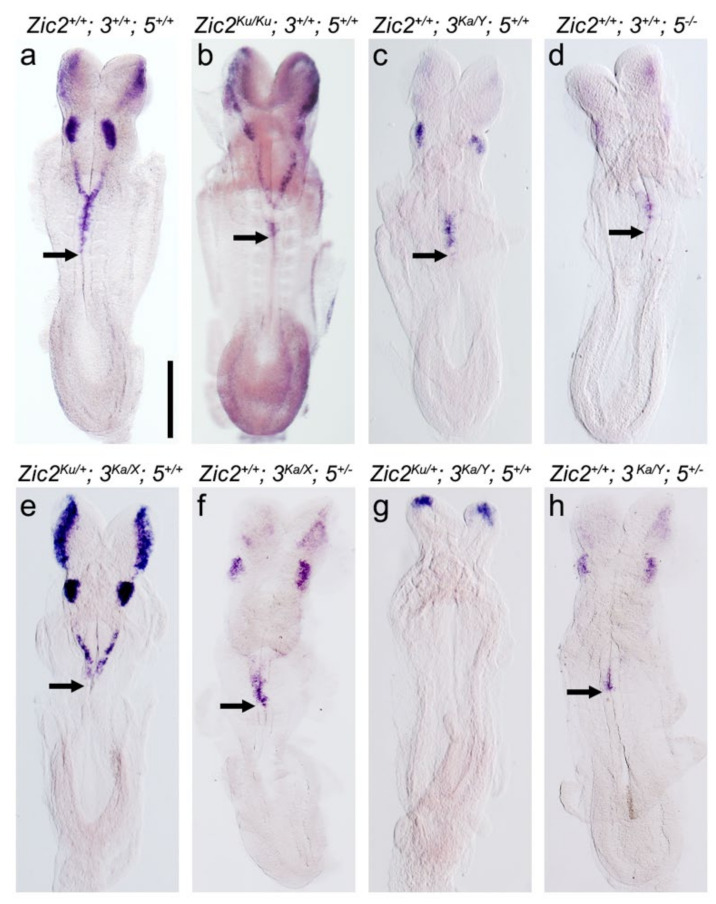
The *Zic* genes co-operate during murine neural crest cell specification. (**a**–**h**) Dorsal view of 8-somite stage murine embryos of the genotypes shown, following WMISH to *Foxd3* (anterior to the top). The black arrow marks the distal extent of *Foxd3* expression. The genotype is shown at the top of each panel and is abbreviated such that the *Zic* gene symbol is shown only once per panel. The murine *Zic* genes are located as follows (assembly GRCm39) *Zic5*—Mmu 14: (122696572..122703127, complement), *Zic2*—MMu 14: (122712796..122717740), and *Zic3*—MMu X: (57075988..57081990) and would normally be in chromosome order in a compound genotype. Instead, they are written in numerical order for ease of reading (i.e., in panel a, the genotype *Zic2^+/+^*; *3^+/+^*; *5^+/+^* is used to represent *Zic5^+/+^*; *Zic2^+/+^*; *Zic3^+/+^*). Scale, 200 μm.

**Figure 3 ijms-22-10437-f003:**
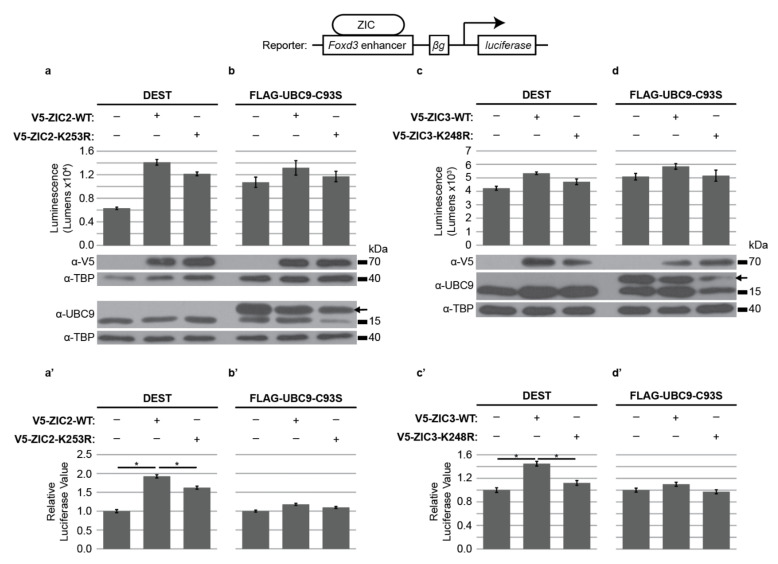
SUMOylation increases ZIC2 and ZIC3 transactivation of the *Foxd3* enhancer. Luciferase activity at the *Foxd3* reporter in HEK293T cells. (**a**,**a**’) The K253R SUMO-incompetent form of ZIC2 shows a significant decrease in transactivation ability compared to wild-type ZIC2. (**b**,**b**’) The transactivation ability of wild-type ZIC2 is impeded when SUMOylation is universally inhibited via UBC9-C39S and is equivalent to the SUMO-incompetent form of ZIC2. (**c**,**c**’) The K248R SUMO-incompetent form of ZIC3 shows a significant decrease in transactivation ability compared to wild-type ZIC3. (**d**,**d**’) The transactivation ability of wild-type ZIC3 is impeded when SUMOylation is universally inhibited via UBC9-C39S and is equivalent to the SUMO-incompetent form of ZIC3. (**a**–**d**) Raw data and WB showing overexpressed proteins from one representative experiment. For WB of nuclear fractions (using antibodies against V5 or UBC9) antibody against TBP was used as a loading control. The arrow in the UBC9 WB denotes the larger, tagged exogenous protein. In (**a**,**b**), the ZIC and UBC9 proteins were detected on separate blots, whereas in c and d all proteins were detected on one blot. Error bars denote ±s.d. from three internal repeats. (**a**’–**d**’) Pooled data from three external repeats (normalized to background). Error bars denote ±s.e.m. *: *p* < 0.05, two way ANOVA with Bonferroni multiple comparison test.

**Figure 4 ijms-22-10437-f004:**
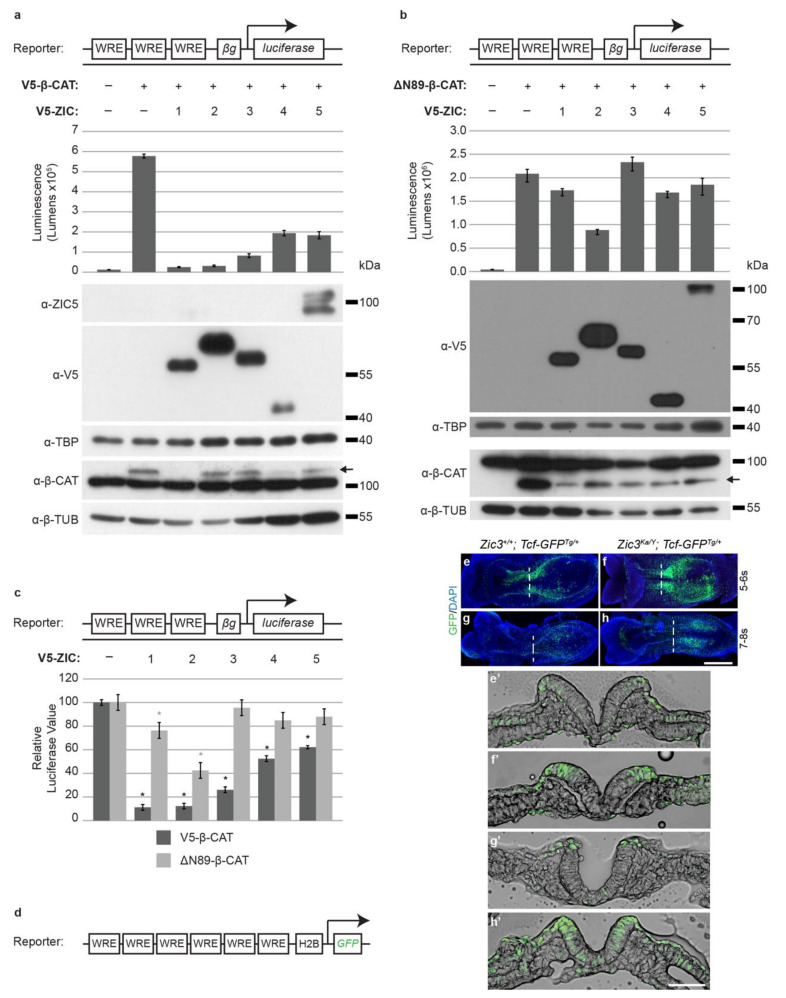
Context-dependent inhibition of WNT/β-catenin-mediated transcription by the ZIC proteins. (**a**–**c**) Luciferase activity in HEK293T cells of the ZIC-optimized TCF-reporter construct shown driven by expression of epitope-tagged ZIC proteins (V5-ZIC1-5). WNT signaling was activated by transfection with (**a**) wild-type β-catenin (V5-β-CAT) or (**b**) stabilized β-catenin (ΔN89-β-CAT) to give higher activity. (**a** and **b**) Raw data and corresponding WBs of the overexpressed proteins from one representative experiment. Error bars denote ±s.d. of three internal repeats. For WB of nuclear fractions (using anti-V5 antibodies or anti-ZIC5-serum), antibodies against TBP were used as a loading control. For WB of cytoplasmic fractions (using anti-β-catenin antibody; α-β-CAT) antibody against β-tubulin (α-β-TUB) was used as a loading control. The arrow in the β-CAT WB denotes the larger, tagged exogenous protein. Exogenous β-CAT is depleted to varying extents in the presence of the different ZIC proteins, which is consistent with the observation that ZIC3 overexpression enhances β-catenin degradation [46,56]. (**c**) Pooled data from the experiments shown in a and b presented relative to the value given by the respective empty-vector control transfections. In each data set the relative luciferase value of the β-catenin transfection is set to 100%. Error bars denote ±s.e.m. of three external repeats, *: *p* ˂ 0.05, two-way ANOVA with Fischer’s unprotected post ad hoc test. Black asterisk, V5-β-CAT data; gray asterisk, ΔN89-β-CAT data. Statistics to compare V5-β-CAT data and ΔN89-β-CAT data were not performed. (**d**) Line diagram of the TCF/Lef:H2B-GFP mouse strain transgenic construct. (**e**,**f**,**g**,**h**) Dorsal view of wholemount embryos showing GFP (green) reporting TCF-mediated transcription and the DAPI (blue) nuclear stain. Scale, 200 μm. (**e**’,**f**’,**g**’,**h**’) Transverse sections at levels shown in corresponding image of embryos, showing GFP (green) reporting TCF-mediated transcription and a bright-field image of the section. The embryos’ genotype and stage are shown on each panel. Scale, 100 μm.

**Figure 5 ijms-22-10437-f005:**
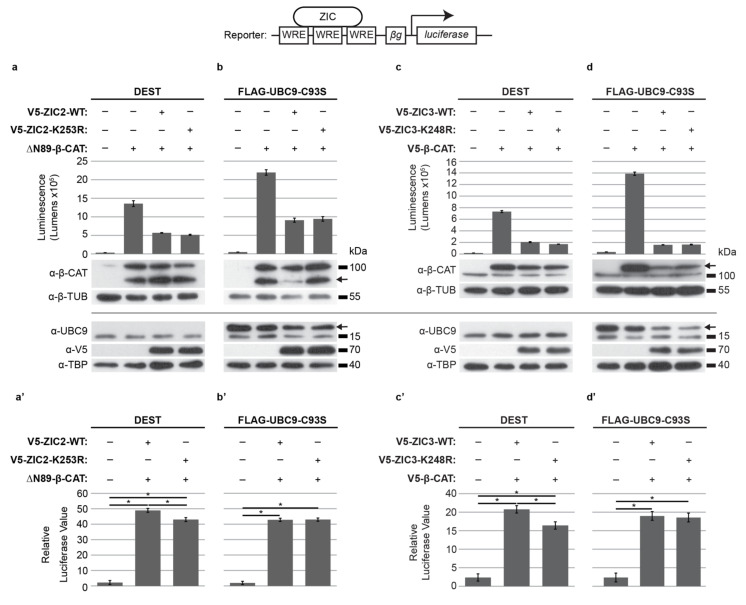
SUMOylation reduces ZIC inhibition of WNT signaling. Luciferase activity at the ZIC-optimized WRE reporter in HEK293T cells. (**a**,**a**’) The K253R SUMO-incompetent form of ZIC2 shows a significant increase in inhibition of β-catenin-mediated transcription compared to wild-type ZIC2. (**b**,**b**’) The ability of wild-type ZIC2 to inhibit β-catenin-mediated transcription is increased when SUMOylation is universally inhibited via UBC9-C39S and is equivalent to the SUMO-incompetent form of ZIC2. (**c**,**c**’) The K248R SUMO-incompetent form of ZIC3 shows a significant increase in inhibition of β-catenin-mediated transcription compared to wild-type ZIC3. (**d**,**d**’) The ability of wild-type ZIC3 to inhibit β-catenin-mediated transcription is increased when SUMOylation is universally inhibited via UBC9-C39S and is equivalent to the SUMO-incompetent form of ZIC3. (**a**–**d**) Raw data and WB showing overexpressed proteins from one representative experiment. For WB of cytoplasmic fractions (using antibody against β-catenin; α-β-CAT), antibody against β-tubulin (α-β-TUB) was used as a loading control. The arrow in the α-β-CAT WB (**a**–**c**) denotes the tagged exogenous protein (V5-β-CAT is larger and ΔN89-β-CAT is smaller than the endogenous β-CAT). For WB of nuclear fractions (using antibodies against V5 or UBC9) antibody against TBP was used as a loading control. The arrow in the UBC9 WB denotes the larger, tagged exogenous protein. Error bars denote ±s.d. from three internal repeats. (**a**’–**d**’) Pooled data from three external repeats (normalized to background). Error bars denote ±s.e.m. *: *p* < 0.05, two-way ANOVA with Bonferroni multiple comparison test.

**Figure 6 ijms-22-10437-f006:**
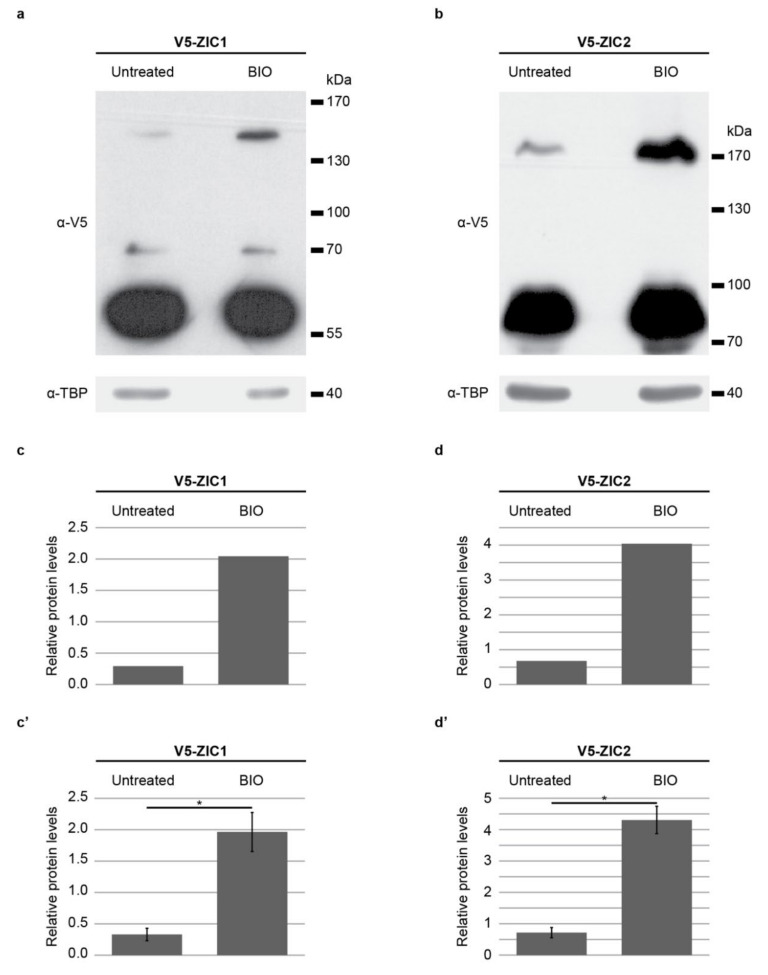
Activation of the canonical WNT pathway promotes ZIC SUMOylation. (**a**–**d**’) Quantification of SUMOylated ZIC based on WB analysis of ZIC protein in the nuclear fraction of HEK293T cells co-transfected with V5 epitope-tagged ZIC expression plasmids and EmGFP-SUMO1-WT followed by incubation in the presence or absence of the GSK inhibitor (2′Z,3′E)-6-Bromoindirubin-3′-oxime (BIO). (**a**,**b**) Representative WBs for the ZIC1 or ZIC2 analysis. For WB of nuclear fractions (using antibody against V5), antibody against TBP was used as a loading control. (**c**,**d**) Quantification of the SUMOylated ZIC protein in the presence or absence of BIO in the WB shown in (**a**,**b**), respectively. ZIC1 and ZIC2 protein levels were quantified relative to TBP using ImageJ. (**c**’,**d**’) Average level of SUMOylated ZIC protein from five independent experiments. For both ZIC1 and ZIC2, incubation with BIO significantly increases the amount of the SUMOylated form of ZIC. *: *p* < 0.01 two-sample *t*-test. Error bars denote ±s.e.m.

**Figure 7 ijms-22-10437-f007:**
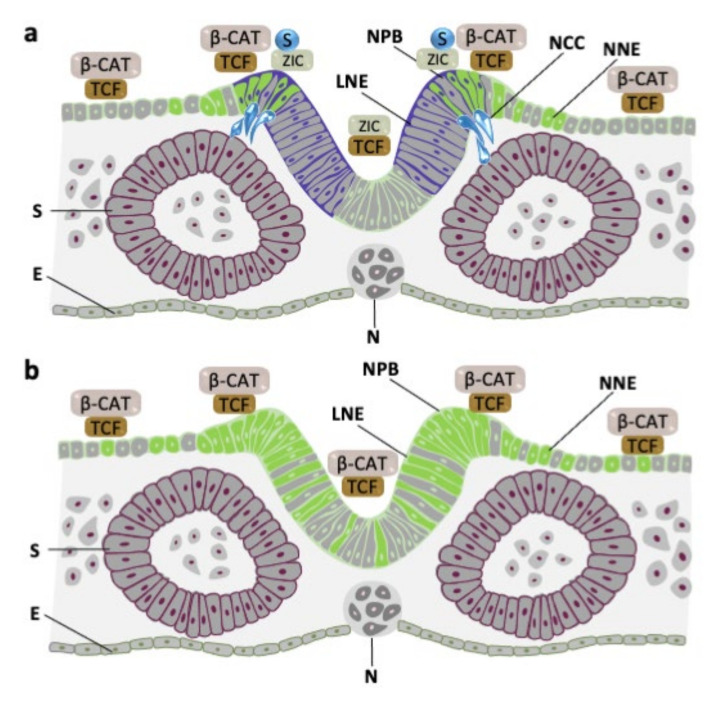
SUMOylation of ZIC proteins can influence NCC specification. A diagram explaining how ZIC protein activity at the neural plate border can restrict the region of NCC specification. (**a**) During normal development, WNT signaling sufficient to cause TCF activation of gene expression is restricted to the neural plate border and some non-neural ectoderm cells. *Zic* gene expression is within the lateral neurectoderm and neural plate border. In the lateral neurectoderm, basal ZIC acts as a TCF co-repressor. At the neural plate border, high-WNT activity promotes ZIC SUMOylation and transcription factor activity at ZREs, including those that drive expression of the *Foxd3* specifier gene. (**b**) When *Zic* gene expression is depleted in the lateral neurectoderm, TCF activation of gene expression expands throughout the neurectoderm and into the non-neural ectoderm. Simultaneously, the lack of ZIC protein prevents activation of *Foxd3* expression. E: endoderm, LNE: lateral neurectoderm, NCC: neural crest cells, NPB: neural plate border, NNE: non-neural ectoderm, N: notochord, S: somite.

## Data Availability

The data presented in this study are available on request from the corresponding author.

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
