# Peer review of "SUMOylation Potentiates ZIC Protein Activity to Influence Murine Neural Crest Cell Specification"

_ijms, 2021, doi:10.3390/ijms221910437_

Round 1
Reviewer 1 Report
This is an interesting manuscript studying the role of SUMOylation of different ZIC transcription factors in NCC specification. These are my comments:
- Have the authors used anti-SUMO antibody to detect the endogenous level of ZIC proteins after co-IP? For the overexpressed SUMOylation experiments, anti-SUMO antibody can be used in parallel to confirm that the high MW bands are SUMOlated proteins.
- For Zic 2, 3, 5 compound mutants, were later stages of embryos studied? Were there any developmental defects? Was the migration of NCC affected in these mutants?
- In addition to Foxd3 as a maker gene of NCC specification, can authors use another gene to verify some of the findings?
Reviewer 2 Report
In the manuscript, Bellchambers et al. revealed the role of SUMOylation of the ZIC proteins in neural crest induction. Overall, I think it is a good manuscript that could be published in IJMS. However, there are a few experiments that could be done to make the manuscript even better:
- It would be more convincing if the authors could show Zic2, Zic3, and Zic5 are expressed at the neural plate border during neural crest specification.
- Our unpublished data show that even the expression of Foxd3 and Sox10 is disrupted, neural crest cells could still be induced and able to migrate to their target region. Therefore, other neural crest genes, such as Pax3, Ap2, Msx1, Msx2, Notch1, and Twist, should be tested to confirm neural crest specification is disrupted in Zic protein mutant embryos.
- Foxd3 is known to maintain the stem cell potential of neural crest cells. Therefore, are there any neural crest-related phenotypes in Zic protein mutant embryos?
